# An evaluation of Roluperidone as a promising repurposing candidate for Alzheimer's Disease: A Computational Investigation

Rehnuma Tanjin[1], Md. Al-Amin [1]*, Jannatul Mawa Etee[1], Ayesha Siddika[1], Ahmadullah Siddiki[2], Saiful Islam Mahi[3], Sharmin Nur Toma[1], Nafisa Akter[1], Md. Helal Uddin[1], Neelima Akhter Bristy[1], Samira Idris Mowlee[1], Elmu Kabir Rafa[1], Md. Faruk Hossen[1]

**1** Department of Pharmacy, Islamic University, Kushtia, Bangladesh, **2** Applied Chemistry and Chemical Engineering, Islamic University, Kushtia, Bangladesh, **3** Department of Biomedical Engineering, Islamic University, Kushtia, Bangladesh

* alamin1730021@gmail.com

## Abstract

Alzheimer's disease (AD) is the most dominant and prevalent form of dementia. The therapeutic agents for AD are not sufficient. Drug repurposing (*i.e.,* also called drug repositioning or therapeutic switching of drugs) could contribute to adding novel therapeutic agents in AD discovery pipeline. Blood-brain barrier (BBB) is a crucial factor, for brain's diseases related drug discovery. Since, CNS active compounds have BBB crossing property, in this study this category of compounds was re-evaluated as repurposing potential candidate for AD by integrated machine learning algorithm, cheminformatics analysis, molecular Docking and simulation-based approach. We built three machine learning model such as Support Vector Machine (SVM), Random Forest (RF), Extreme Gradient Boosting (XGB) for the prediction of AD potential repurposing candidates. The SVM classification model performed better than others. The SVM classification model achieved an Area Under the Curve of the Receiver Operating Characteristics (ROC-AUC) of 0.81, along with higher precision, recall, and F1 scores. The support vector machine (SVM) was implemented to classify 500 CNS active compounds as AD drug potential and non-AD drug potential. Using the SVM model, 60 compounds were predicted as AD repurposing potential from 500 CNS active compounds. Structural similarity analysis of 60 compounds with Donepezil as a reference drug was performed using 5 different types of fingerprints such as 'substructure', 'extended', 'circular', 'EState', 'MACCS'. 9 compounds from them obtained as structurally most similar to the reference drug. After the molecular docking performance of 9 compounds into the active site & peripheral anionic site of human acetylcholinesterase (hAChE), it was revealed that Roluperidone' had binding affinity of −12 kcal/mol, and 'Napitane' had binding affinity of −11.9 kcal/mol whereas the reference drug Donepezil had a binding affinity of −11.8 Kcal/mol. Molecular dynamics

**Data availability statement:** All relevant data are within the manuscript and its Supporting Information files.

**Funding:** The author(s) received no specific funding for this work.

**Competing interests:** The authors have declared that no competing interests exist.

**Abbreviations:** AD, Alzheimer's Disease; CNS, Central Nervous System; SPP, Similarity Property Principal; ML, Machine Learning; SVM, Support Vector Machine; RF, Random Forest; AChE, Acetylcholinesterase; hAChE, Human Acetylcholinesterase; ML, Machine Learning; ROC-AUC curve, Receiver Operating Characteristic-Area Under curve; LVS, ligand-based virtual screening; MACCS, Molecular Access System; PAS, Peripheral Anionic Site; ADHD, Attention Deficient Hyperactivity Disorder; CAS, Catalytic anionic Site; MDS, Molecular Dynamic Simulation; RMSD, Root Means Square Deviation; Rg, Radius of Gyration; RMSF, Root Means Square Fluctuation

simulation revealed that Roluperionde had better binding integrity to hAChE. This study laid out computational reinvestigation of 500 CNS active drugs for therapeutic switching to AD, and 'Roluperidone' is found as an AD repurposing potential candidate. However, *in-vitro* and *in-vivo* studies are further needed to fully elucidate the compound's potential as AD repurposing drugs.

## 1. Introduction

Globally, dementia is identified as the fifth leading cause of death [1]. Worldwide 55 million people have suffered from dementia, more than 60% of whom live in low-and middle-income countries, and approximately 10 million new cases of dementia per year. AD is a neurodegenerative disease. The nerve cells related to thinking and memorizing are severely damaged by AD [2].

Fig 1 represents that in the United States of America, 6.07 million older adult AD patients were estimated in 2020 and it will be increased by 7.16 million in 2025 to 13.85 million in 2060 [3].

The pathologic hallmarks of AD are senile plaque and neurofibrillary tangles (NFT). Beta and gamma-secretases (*i.e.,* membrane protease enzymes) break down ß-amyloid precursor protein, as a result, amyloid beta protein is produced [4]. Amyloid beta protein comprises senile plaque. Hyperphosphorylated tau (*i.e.,* a microtubule-associated protein) is a dominant constituent of neurofibrillary tangles. In the brain's area where NFT is formed, neuronal loss has occurred in AD [5–8].

To date on January 1, 2023, there were 36 drugs in phase 3 of clinical trials for AD [9]. There is not a sufficient quantity of drugs in AD drug discovery pipeline to battle against this disease. Drug repurposing (*i.e.,* it is a process that reinvestigates the marketed drugs as well as clinical trial drugs for novel therapeutic applications) could enhance the AD drug discovery pipeline [10]. 'Galanthamine' (*i.e.,* an acetylcholinesterase inhibitor)' was repurposed for AD. Early it was used to treat poliomyelitis [11]. The drug repurposing process for AD could salvage the drug development time as well as cost (*i.e.,* based on the therapy type or developing firm, the cost of a new drug candidate in a clinical trial is around $500 million to more than $2,000 million and the time would be required up to 20 years for FDA approval of a new drugs candidate) [11–13].

Virtual screening is used to identify novel bioactive compounds. Usually, ligand-based and structure-based methods are implemented in virtual screening. There are various approaches for ligand-based virtual screening, such as screening of compounds (1) by machine learning, (2) by fingerprint similarity method, (3) by shape-based similarity, *etc* [14]. Here, we implement both ligand-based (i.e., similarity searching and machine learning) and structure-based (i.e., molecular docking & simulation) methods for virtual screening of CNS active compounds.

Several computational studies have been performed for drug repurposing for AD. For example, Shivani et al. screened 150 antipsychotic drugs by structure-based virtual screening (*i.e.,* molecular docking) for therapeutic switching to AD [15]. Steve

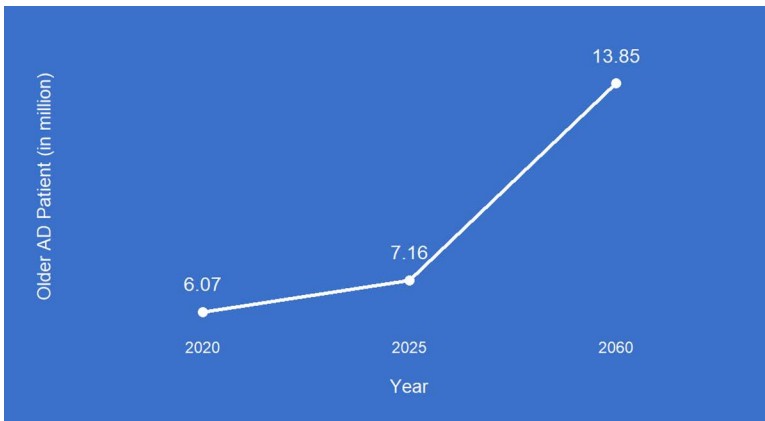

**Fig 1. Older AD patient statistics in the USA.**

et al. described a machine-learning model using gene and neural cell culture data to identify novel repurposing candidates [16]. G N S et al. described protein-protein interaction network, homology modelling, molecular docking, dynamic based approach for AD drug repurposing [17]. Kumar et al. performed molecular docking and bioassay and obtained thiazolidine-dione (TZD, antidiabetic) and aminoquinoline (antimalarial) as potential repurposing candidate for AD [18].

So far, to the best of our knowledge, there has been no study based on a combination of machine learning, cheminformatics, molecular docking & simulation for AD drug repurposing. Moreover, collectively all the CNS disease-related drugs were not chosen for AD drug repurposing. In this study, we aim at training learning (ML) algorithms along with cheminformatics analysis in term of structural similarity searching, molecular docking, and molecular dynamics simulation to predict AD repurposing potential repurposing candidate from 500 CNS active compounds. Our study suggests that 'Roluperidone' has the potential to be a novel repurposing candidate for AD. Here, the interactions of 'Roluperidone' with hAChE is depicted in Fig 2.

From Fig 2, it is seen that Roluperidone interacts with the active site residues of hAChE that are investigated computationally. However, solely computational work is not enough. So, *in-vitro* & *in-vivo* experiments have to be performed to justify the study.

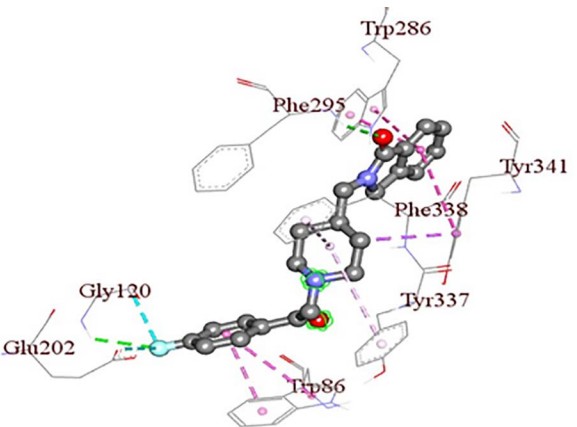

**Fig 2. The interactions between Roluperidone & hAChE.**

## 2. Materials and methods

### 2.1 Repurposing targeting compounds enlistment

We chose the central nervous system (CNS) related diseases (*i.e.,* except AD) such as epilepsy, Parkinson's disease, insomnia, depression, migraine prophylaxis, attention deficit hyperactivity disorder (ADHD), schizophrenia, bipolar I disorder, Tourette's disorder, anxiety disorder, obsessive-compulsive disorder (OCD), lennox-gastaut syndrome or dravet syndrome, seasonal affective disorder, multiple sclerosis, *etc* for repurposing-targeting compounds selection. For CNS diseases relevant drug discovery, the BBB permeability is an important factor [19]. The prescribed drugs or investigated compounds for these diseases have the property of crossing the blood-brain barrier (BBB). Since these drugs can cross BBB, we hypothesized that they may had repurposing potential to AD. In total of 500 compounds comprising 320 approved & 180 investigational CNS active drugs related to the above-mentioned diseases were selected by exploring two databases: (1) 'Drug Bank' (2) 'NCATS Inxight Drugs' for elucidating their repurposing potential for AD [20,21]. This enlisted file was given as a piece of supporting information (S1 File. AD Repurposing Candidate Enlistment.xlsx).

### 2.2 Machine learning (ML)-based classification models

**2.2.1 Molecular descriptors extraction & dataset preparation for ML models.** Some bioassay data (i.e., these bioassay data regarding to the compounds' inhibitory activity testing against acetylcholinesterase) were chosen for the machine learning model building. The dataset for machine learning model building contained in total of 615 compounds. The molecular descriptors for these compounds were calculated through Padel software. 1475 descriptors which included (i.e., XLogP, Rotatable Bonds Count, WHIM Descriptor, Fractional PSA Descriptor, *etc.)* were calculated for each compound that was given as a piece of a supporting information [S2 File. Training Building Dataset.xlsx]. The python-based library called Scikit-learn was implemented to build machine learning model. The codes used for model building is given as a piece of supporting information [S3 File. Python script for machine learning model generation.docx]. Initially the dataset was preprocessed before machine learning model building. The preprocessing steps were (a) replacing the nan value by the median of the respective variable; (b) standardization of the data. Then the most regarding features to the target variable were chosen by setting the p value 0.05. The p-value equal or less than 0.05 is statistical importance [22]. Here the 'Anova F' method was implemented to select the most crucial features basis on the p-value. Then the whole dataset was divided into 3 section such as training (80%), testing (10%), and external (10%). The training dataset was used to train the model, the testing dataset was used to internal validation of the model, and the external dataset for the external validation of the model [30].The training set contained both active & inactive compounds data as well as the test set. For this reason, subsequently, they could detect which compounds that would be active for repurposing potential and which would be non-active from the prediction dataset.

**2.2.2 Machine learning algorithms.** In ligand-based virtual screening (LVS), the machine Learning (ML) algorithm is an attractive technique for the classification of compounds. ML algorithm is applied to classify the compound as active and inactive based on the model derived from the training dataset [23]. In drug repurposing, machine learning (ML) methods such as random forest (RF), support vector machine (SVM), and k-nearest neighbors play important roles [24]. Moreover, ML algorithms improve in the small-molecule design, understanding of disease and non-disease phenotypes, *etc* [25]. Support Vector Machine (SVM) is a statistical method that maps data into high-dimensional space to recognize a lower-dimensional hyperplane that maximizes the data segregation utilizing nonlinear kernel and it is gained by maximizing the margins between hyperplanes commonly known as support vectors [26]. Random forest is a classification technique. It based on multiples decision trees and voting rules [27]. Here, RF, SVM, & extreme gradient boosting (XGB) were trained using 5-fold cross-validation. Then we evaluated the models performance using internal validation set as well as external validation set.

**2.2.3 ML model assessment.** The SVM and RF model's performance were evaluated base on AUC-ROC curve, accuracy, precision, recall, and f1 scores. True positive (TP) or sensitivity is the ratio of precisely classified true active (*i.e.,* here the number of AD compounds accurately determined as positive). True negative (TN) or specificity is the ratio of appropriately classified true negative (*i.e.,* here the number of decoy compounds correctly determined as negative). False positive (FP) is the number of decoy compounds that are misclassified in terms of decoys that are determined as active although they are not active. The number of AD compounds that are inappropriately classified as inactive was defined as false-negative (FN). The area under the curve (AUC) measures the true positive rate versus the false positive rate. The overall performance of the ML models were assessed by the accuracy, precision, recall, and F1 scores. These can be presented by the following expressions:

$$Accuracy = \frac{(TP + TN)}{(TP + TN + FP + FN)} \tag{1}$$

$$Precision = \frac{TP}{TP + FP} \tag{2}$$

$$Recall = \frac{TP}{TP + FN} \tag{3}$$

$$F1\ Scores = 2 \times \frac{Precision \times recall}{Precision + recall} \tag{4}$$

Here, TP = true positive, TN = true negative, FP = false positive, FN = false negative

### 2.3. Cheminformatics analysis

**2.3.1 Calculation of compound's fingerprints.** Five different types of fingerprints: 'substructure', 'extended', 'circular', 'MACCS', E-State' were calculated for each compound using the cheminformatics toolkit 'rcdk'. The general process for calculating fingerprints by rcdk involves several steps: first, structural data files (SDFs) were imported into the toolkit. Next, inherent numerical features, such as molecular weight and LogP, were used to generate a unique binary representation (a fingerprint) for each molecule. These binary fingerprints were then quantitatively compared using statistical approaches, like the Tanimoto coefficient, to compare molecular similarity. The R script used for fingerprint calculations is provided in the supporting information.[S4 File. Rcdk script for fingerprint calculation.txt]

The 'atom-pair' fingerprint is crucial for similarity searching, lead hopping, and structure-activity relationships [28–30]. The 'extended' fingerprint is a topological fingerprint that carries stereochemical features as well as sub-structure information [31]. E-State is a 79-bit length-based structural fragment fingerprint [32]. Since each fingerprint contains a specific structural description of a compound, it would be more realistic to use several fingerprints together for similarity calculation.

**2.3.2 Structural similarity calculation.** Similarity searching of compounds is a widely used technique in pharmaceutical research to screen compounds [33]. This technique has many advantages. For example, structurally similar compounds may be more active than randomly picked compounds [34].

The three principal components are involved in similarity measurements of chemical compounds such as (i) similarity coefficient; (ii) structural descriptors, and (iii) weighing schemes. The similarity coefficient like Tanimoto coefficient is expressed by the following equation:

$$SA, B = \frac{c}{a + b - c} \tag{5}$$

Here, S represents the 'Tanimoto Coefficient', a = the number of unique fragments in the compound 'A' & b = the number of unique fragments in the compound 'B'. c = the number of unique fragments shared by compounds A and B [35]. The cheminformatics toolkits 'rcdk' & 'ChemmineR' was used for the similarity calculation [36]. Donepezil was chosen as a reference drug for similarity calculation because it is one of the first-choice drugs for AD treatment [37]. The calculated similarity scores were given as a piece of supporting information [S5 File. Similarity scores.xlsx]

**2.4 Protein & ligands preparation**

The X-ray crystallographic structure of the human acetylcholinesterase (hAChE) complex with Donepezil (PDB ID:4EY7) was retrieved from the Protein Data Bank (PDB) (https://www.rcsb.org/) database [38]. 'BIOVIA Discovery Studio Visualizer' was used to clean the protein-ligand complex by extracting the co-crystallized ligand & water molecules. The energy minimization of the protein was performed by the Swiss-PDB Viewer. The required ligands for molecular docking were downloaded from the PubChem database.

**2.5 Molecular docking**

Molecular Docking was performed by MGL software of 'Autodock-Vina' [39] to figure out all the possible orientations and conformations of the ligands with peripheral anionic sites (PAS) and catalytic anionic sites (CAS) of the active site pocket of hAChE. The grid box was centered on the active site of the hAChE, which was typically defined based on the co-crystallized ligand that infers the active side residues. The center of the grid box was set at x = −13.81, y = −43.9, and z = 32.76 Å The dimensions of the grid were chosen to adequately cover the entire binding pocket, set to 25x27x25 Å along with the x,y, and z axes, respectively. The binding affinity of the ligands to hAChE was calculated in kcal/mol [40]. As a reference, the binding affinity of the Donepezil was calculated at −11.8 kcal/mol. The interacted amino acids of the hAChE with the ligands were visualized using chimera software. All the data of molecular docking is given as a supporting information [S6 File. Binding Affinity.zip].

**2.6 Molecular dynamics simulation**

The molecular dynamics of three ligand-protein complexes such as Donepezil-hAChE, Napitane-hAChE, Roluperidone-hAChE were performed by YASARA, version 23.9.29.W.64 over a period of 100 nanoseconds with 401 snapshots and the AMBER14 force field [41]. The transferable intermolecular potential3 points (TIP3P) water model was used to solvate the protein-ligand complexes [42]. The physiologic condition was maintained with the addition of Na+ and Cl− ions. All the data of the molecular dynamics simulation is given as supporting information [S7 File. Molecular Dynamic Simulation].

# 3. Results and discussion

### 3.1 Machine learning model-based classification method

In total, 1475 molecular descriptors, including 1D,2D & 3D descriptors, were extracted for the training, testing, external, and predictive datasets. The calculated descriptors are related to constitutional, electrical, hybrid, and geometrical. These descriptors contain unique features of the compounds such the electrical descriptors contain the spatial distribution of electrons, the hybrid descriptors contain the conformational properties of the compounds. The compounds incorporated in the training, testing, and external dataset were collected from PubChem bioassay repository. Based on the $IC_{50}$, values the compounds are classifies as active and inactive. Then ML models - support vector machine (SVM), random forest (RF), and extreme gradient boosting(XGB) were trained using the train dataset and validated by the testing dataset. Several parameters such as accuracy, precision, recall, and F1-score. The evaluation metrics are depicted in Fig 3.

The performance comparison of the RF, SVM, and XGB is evaluated using four classification metrics: Accuracy, Precision, Recall, and F1-score. A higher accuracy score of an ML classification model indicates that the model is good

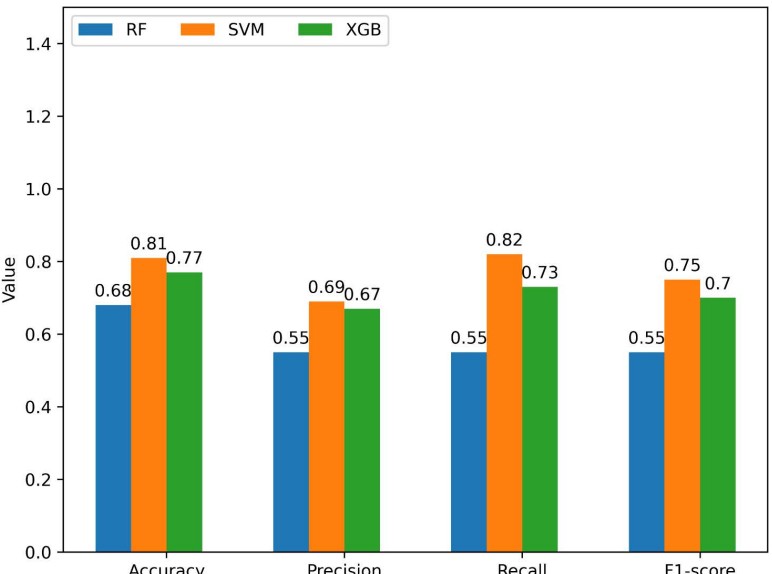

**Fig 3. Performance of the models.**

at making correct classifications. Here, the SVM classification model achieves a comparatively better accuracy (0.81) than the RF and XGB (Fig 3), which indicates that it accurately classifies 81% of the instances. A higher precision value indicates that when the model predicts something is positive, it is most probably to be truly positive. Herein, the SVM classification model shows 69% precision, whereas the RF, XGB model show 55%, and 67% accuracy, respectively (Fig 3). So, the SVM classification performance is better than the others. Furthermore, the SVM classification achieves recall (0.82) and F1scores (0.75) (Fig 3) which are higher than the RF, and XGB classification models. A high recall of a ML model indicates that the model is good at finding all the actual positive instances, and a F1-scores denotes a model has high precision and recall, performing well in accurately identifying positive cases and minimizing both false positives and false negatives. So, it is understood that the SVM model is superior for correctly predicting AD potential repurposing candidates. Moreover, the ML model evaluation also relies on the 'receiver operating characteristic-area under curve (ROC-AUC) curve' analysis [43]. Hence, the SVM, RF, XGB models were also evaluated using the ROC-AUC curve. The following Fig 4, Fig 5, & Fig 6 represent the ROC-AUC curve of the ML models.

From Fig 4, Fig 5, & Fig 6, it is seen that the SVM model has achieved the highest area under curve (AUC) value of 0.81 on the train set. The area under the curve (AUC) value of the RF, XGB model are less than the SVM models. The 'ROC-AUC curve of SVM rises more straightly than the others which in terms of identifying lots of positives without misclassifying lots of negatives. The area under curve (AUC) is directly proportional to the model performance [44]. The SVM model has an AUC value of 0.81, it would be a good classifier [Fig 6]. The AUC value of any ML model greater than 0.80 is an indicator of a good classifier [45]. Moreover, the SVM model was evaluated using an external dataset and it shows approximately a similar result to the obtained result from the training dataset.

### 3.2 Virtual screening of CNS active compounds by support vector machine (SVM) model

The SVM model exhibits better accuracy, precision, recall, and AUC values than the RF, and XGB models. So, it is chosen for the virtual screening for the prediction dataset which contains the information of 500 CNS active drugs. The prediction dataset of 500 CNS active drugs is inputted into the constructed SVM model to predict AD potential repurposing candidates

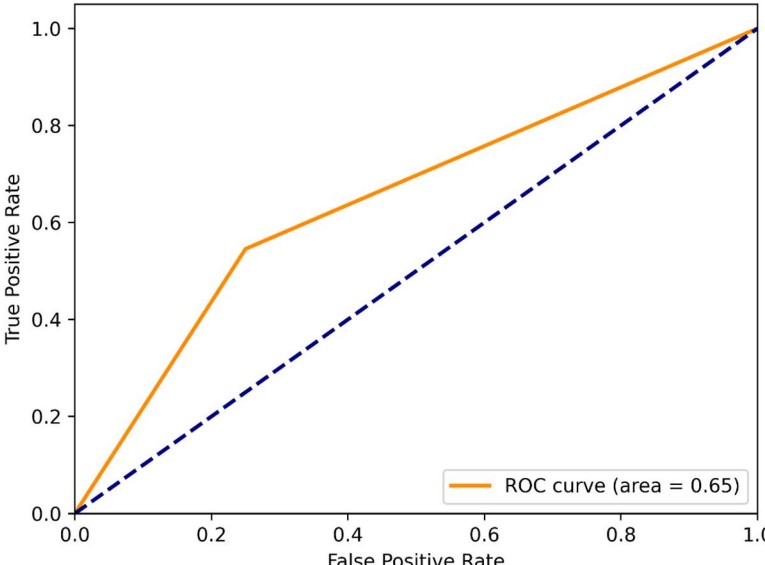

**Fig 4. The ROC-AUC curve of the RF model.**

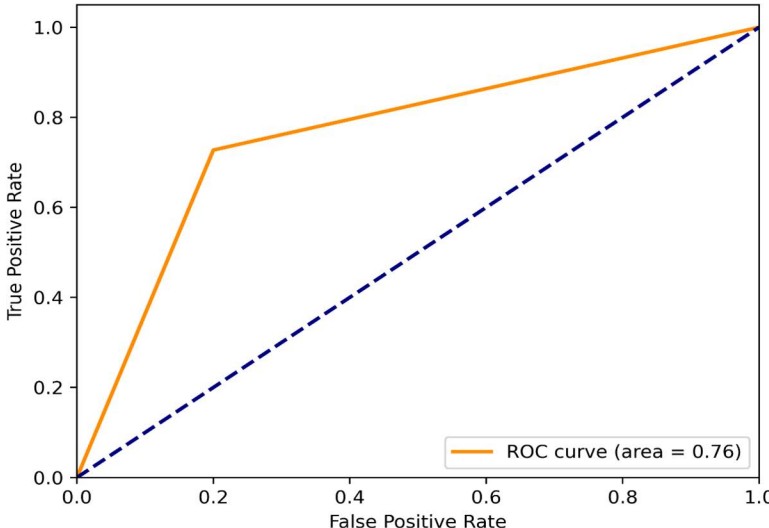

**Fig 5. The ROC-AUC curve of the XGB model.**

and non-AD potential repurposing candidates. Among 500 CNS, drugs 60 compounds are predicted as potential AD potential. In the next step, we perform a structural similarity analysis of 60 AD repurposing potential CNS active compounds which have been screened by the SVM model to select a subset of structurally most similar compounds to Donepezil.

### 3.3 Cheminformatics analysis

**3.3.1 Structural similarity searching.** According to the SPP (Similar Property Principle), "Compounds having structural similarity may have similar therapeutic activity" [46]. SPP is validated by long experience advising rules of

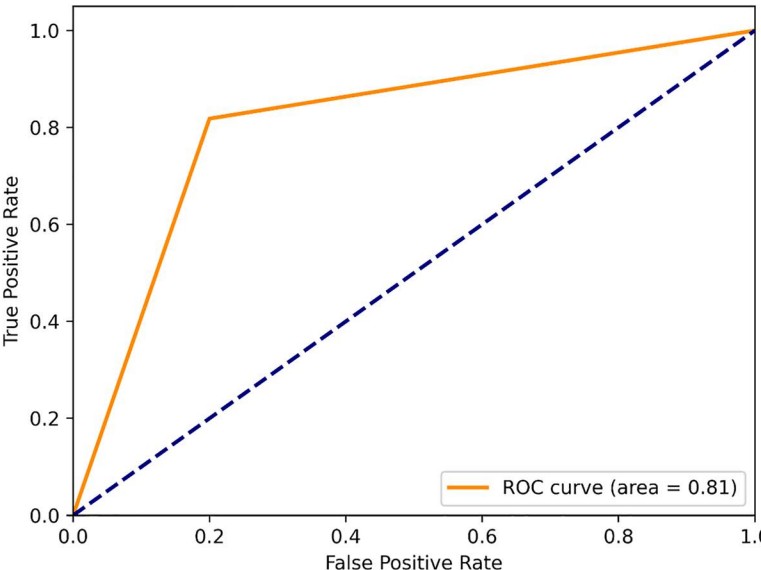

**Fig 6. The ROC-AUC curve of the SVM model.**

thumb such as antibacterial activity in beta-lactam structure, phenethylamine structures involved in CNS (Central Nervous System) activity, and nitro compounds, most probably have mutagenicity [47].Every fingerprint contains various distinct features of chemical structure. Integrating different fingerprints is crucial for the best performance of virtual screening because every fingerprint has unique properties [48]. Therefore, in this study, we combined 5 different fingerprints such as 'substructure', 'extended', 'circular', 'MACCS', E-State' for ligand-based virtual screening of 500 compounds. In this study, we took Donepezil as a standard drug (i.e., a mostly prescribed drug used in mild to severe stages of AD) [49]. Using five of the fingerprints separately, we performed structural-similarity searching of 60 CNS active AD potential compounds (i.e., that had been predicted by SVM model as AD potential) with the reference drug Donepezil. The mean similarity value (i.e., adding every similarity score obtained from 5 different fingerprints, then dividing by 5) for each compound was calculated. We randomly set cut-off of mean similarity value of 0.65. Above this cut-off, 9 compounds from 60 CNS active AD potential compounds are obtained as structurally most similar to the reference drugs. The structural similarity of the 9 compounds with the reference drug Donepezil are shown in the Fig 7.

Fig 7, shows that among 9 compounds 'Rotundine' shows the most structural similarity to Donepezil, having mean similarity scores of 0.7. Then 'Roluperidone' shows higher similarity to Donepezil. 'Ilepcimide' shows the least mean similarity value of 0.64 (Fig 7). The structurally similar molecules are likely to adopt similar binding modes to the same target protein [50]. However, we further proceeded with the 9 CNS active compounds for molecular docking analysis to check whether they have potential binding modes to hAChE or not.

## 3.4 Molecular docking analysis

The successful strategy to slow down Alzheimer's disease (AD), a deteriorating neurological condition, involves suppressing cholinesterase activity [51]. Acetylcholinesterase (AChE) is a serine hydrolase, the enzyme responsible for breaking down the neurotransmitter acetylcholine (ACh) by hydrolysis [52]. Blocking the action of AChE results in an accumulation of ACh concentrations within cholinergic nerve junctions that is essential for cognitive functioning. Therefore, inhibitors of cholinesterase serve as therapeutic agents for a variety of nerve and muscle conditions, like Alzheimer's disease (AD) [53].There is more than one binding site for a ligand in human acetylcholinesterase (hAChE) such as (a) the peripheral

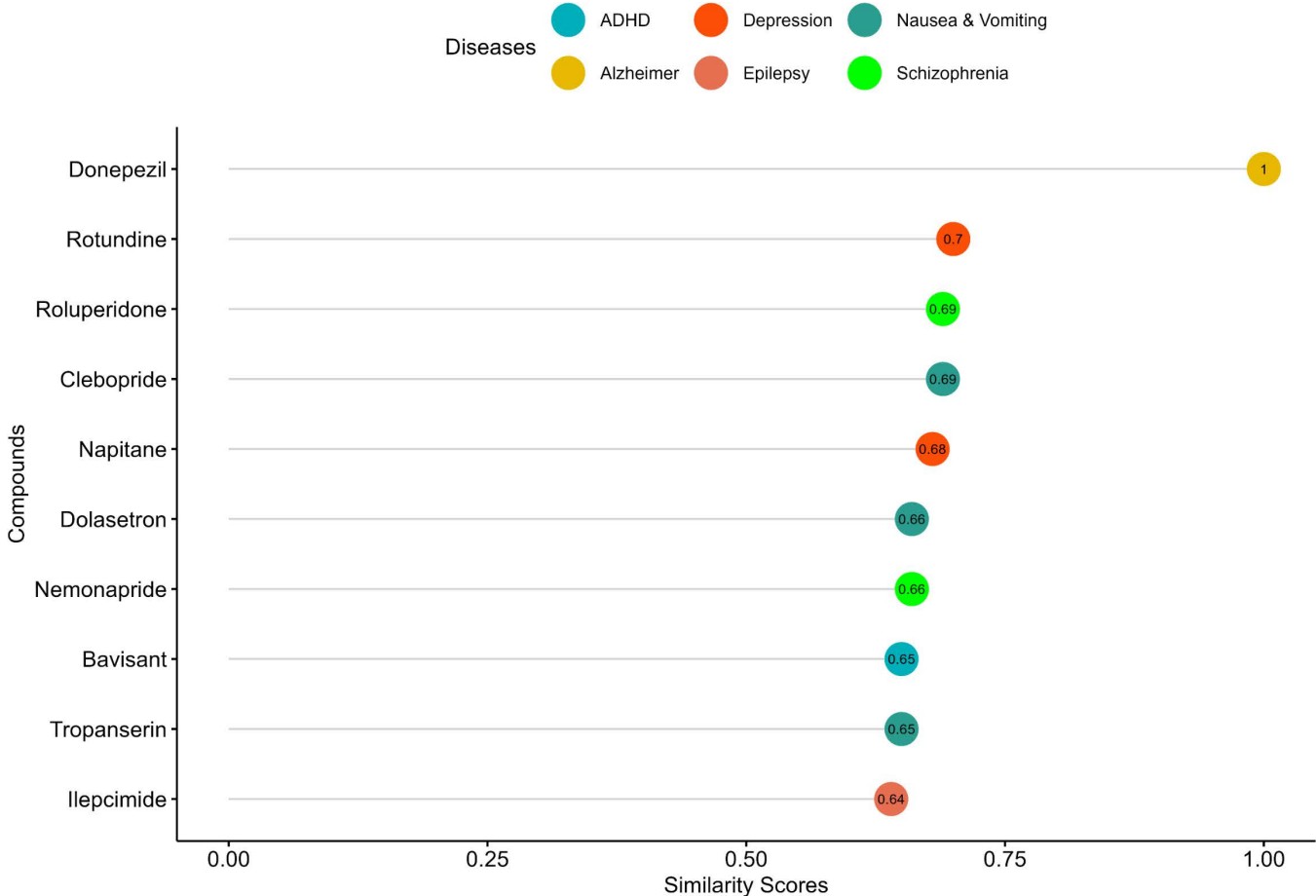

**Fig 7. The Mean Similarity Values of the Top 9 Structural Similar Compounds with the Donepezil.** The Color-marked denoted compounds therapeutic indication.

anionic site (PAS) and (b) the active site's pocket [54]. Furthermore, the active site's pocket of hAChE comprises four sub-pockets such as (i) the catalytic triad (CT) or acylation site or 'A' site which consists of three amino acids residues: Ser203, His447, and Glu334; the A site is responsible for hydrolyzing the ester bond in acetylcholine, (ii) the catalytic anionic site (CAS) is located beside CT; the CAS site made of Trp86, Tyr133, Tyr337, and Phe338; (iii) the acyl binding pocket; it is account for substrate selectivity by preventing excess amount of choline ester entrance; it comprises of Phe295 and Phe297 (iv) and the 'Oxyalion hole' that consists of Gly120, Gly121, and Ala204 [55,56]. The allosteric modulation at the active site of hAChE is controlled by the peripheral anionic site (PAS) which is 18 Å distant from the active site. The PAS site consists of five amino acid residues Tyr72, Asp74, Tyr124, Trp286, and Tyr341 [57]. The 9 CNS active compounds that were obtained as structurally more similar to the reference drug after cheminformatics analysis such as Dolasetron, Clebopride, Rotundine, Tropanserin, Ilepcimide, Nemonapride, Bavisant, Roluperidone, Napitane were underwent molecular docking. Their binding affinity, along with the reference drug, is presented in the Fig 8.

The binding affinity of the reference drug Donepezil was calculated as −11.8 kcal/mol, and the binding affinity of structurally similar 9 CNS active compounds to Donepezil are observed in the range of (−12 to −9.2) kcal/mol (Fig 8). Among 9 CNS active compounds, only two compounds such as '**Roluperidone**', and '**Napitane**' show higher binding affinity than the reference drug seen in Fig 8, it is seen that 'Roluperidone' has a binding affinity of −12 kcal/mol which is greater than

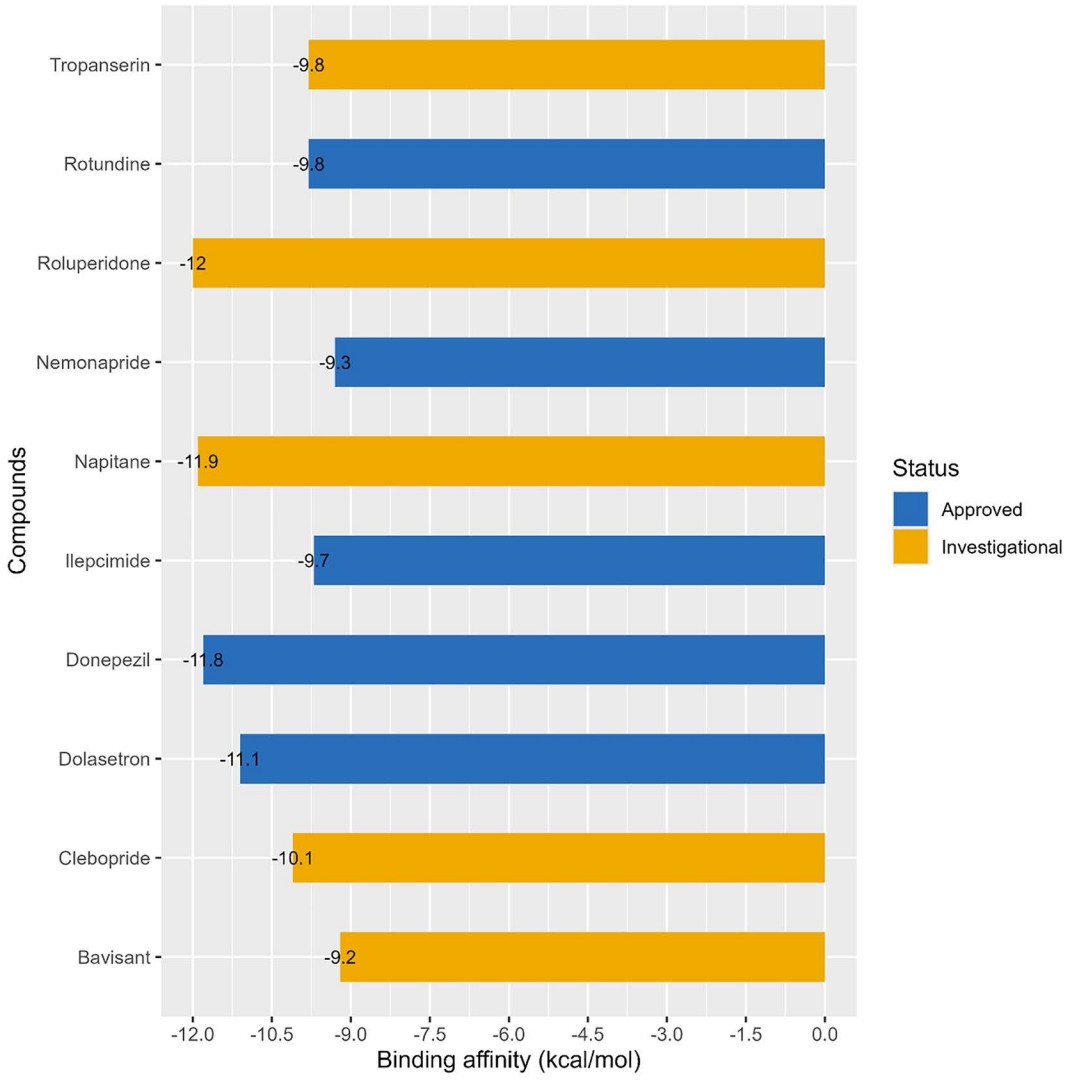

**Fig 8. Binding affinity (kcal/mol) of structurally identical compounds to hAChE.**

among the 9 CNS active compounds as well as the reference drug. 'Napitane' has also a binding affinity of −11.9 kcal/mol (Fig 8). As a therapeutic agent, 'Roluperidone' and 'Napitane' are investigated for schizophrenia and depression respectively. It is seen from Fig 8. that 'Bavisent' has a binding affinity of −9.2 kcal/mol which is the lowest binding affinity among 9 CNS active compounds. The binding interaction of 'Roluperidone', 'Napitane', which have higher binding affinity than the reference drug shown in Fig 9, & Fig 10, Fig 11, and Fig 12, respectively, with the reference drug Donepezil.

From Fig 9, it is seen that Donepezil shows interactions with three PAS site amino acid residues such as Pi-Alkyl bonding with Tyr72, Pi-Sigma bonding with Trp286, and Pi-Alkyl bonding with Tyr341 with a bond length of 4.65227Å, 3.69194Å, 4.83927Å respectively. From Fig 9, Fig 11, & Fig 12, it is seen that 'Roluperidone' and 'Napitane' bind with two PAS site amino acid residues of hAChE such as Trp286, and Tyr341. It forms Pi-Pi stacked interaction with Trp286, Pi-Sigma interaction with Tyr341 (*i.e.,* PAS site residues), having bonding lengths of 3.71588Å, 3.82972Å, respectively (Fig 9). Napitane forms Pi-Pi stacked interaction with both Trp286, and Tyr341(*i.e.,* PAS site residues), having bonding

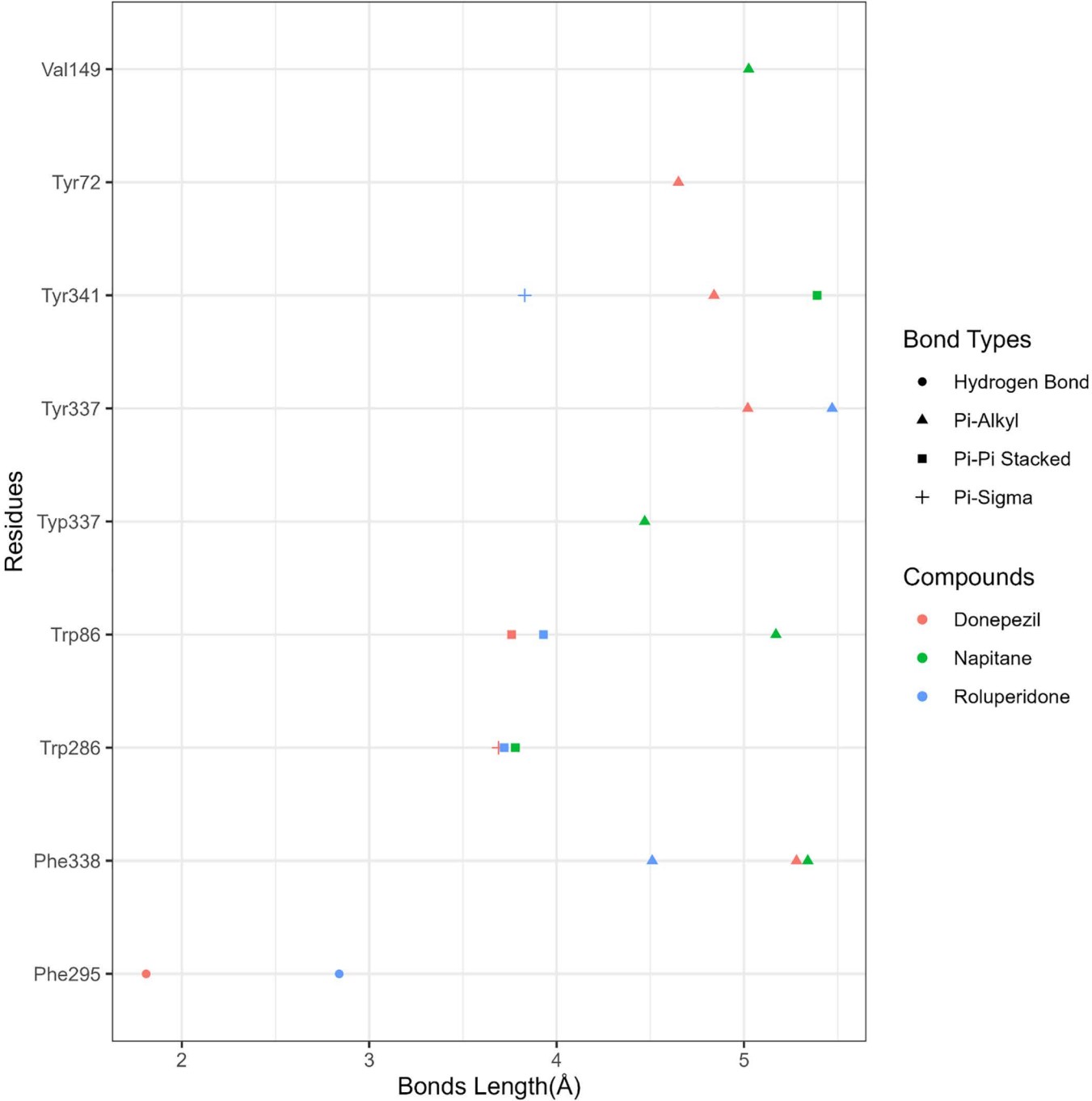

**Fig 9. Bonds length of 'Donepezil', 'Roluperidone', 'Napitane'.**

lengths of 3.7799Å, 5.3891Å (Fig 9). Since amyloid beta accumulation & plaque formation are the main culprits for AD development, the inhibition of aggregated amyloid beta into plaques is an approach to counter AD [58]. hAChE binds with amyloid beta through PAS site amino acid residues and aggregates amyloid beta [59]. However, the interaction of active site amino acid residues is crucial for effectively inhibit the hAChE.

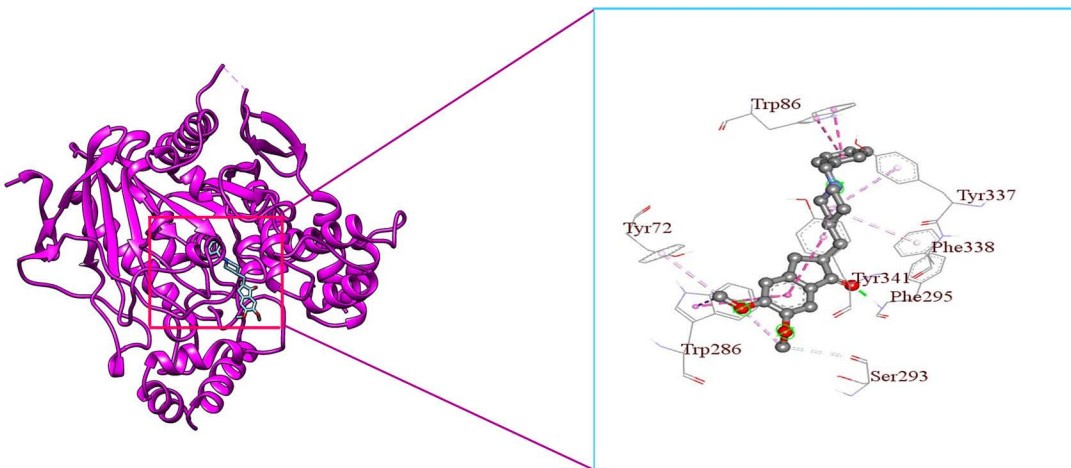

**Fig 10. 3D visualization of Donepezil's binding mode and interaction with hAChE.**

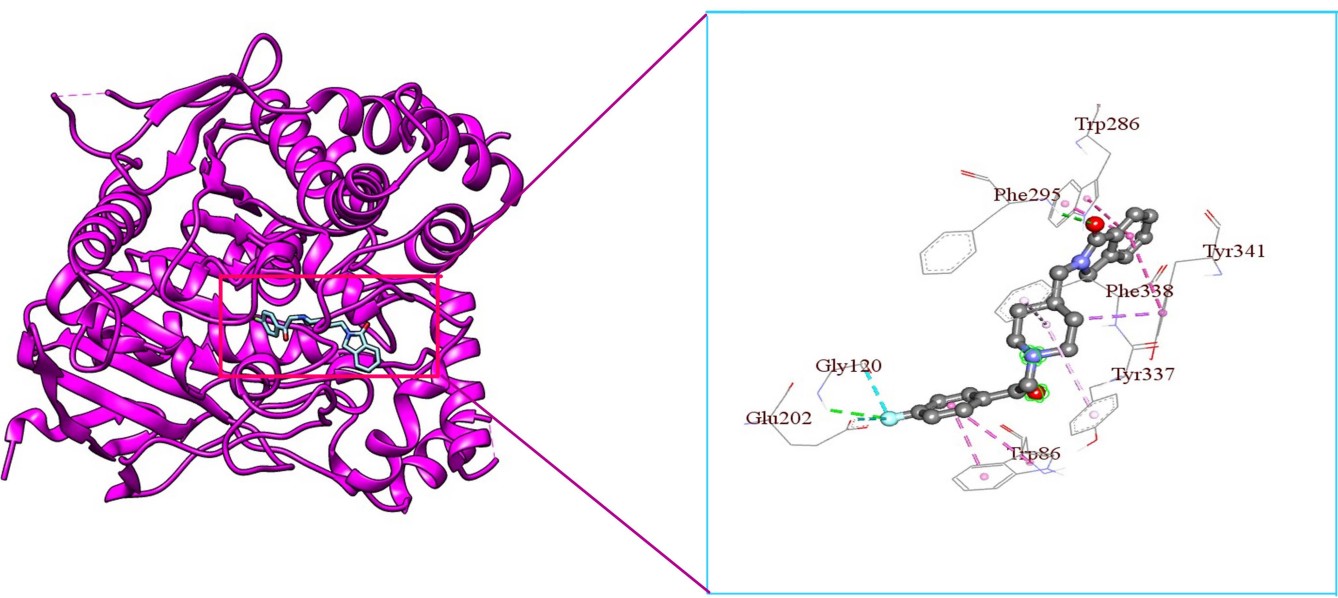

**Fig 11. 3D visualization of Roluperidone's binding mode and interaction with hAChE.**

In animal models of AD, it was found that the rate of amyloid beta production, as well as the amount of amyloid plaque formation, were decreased by the inhibition of both the active site pocket and PAS site of acetylcholinesterase (AChE) [60]. From Fig 9, it is seen that Donepezil binds with three CAS site amino acid residues of active site pocket of hAChE such as Trp86, Tyr337 and Phe338. Likewise, Donepezil, 'Roluperidone' and 'Napitane' bind with CAS site amino acid residues of hAChE which are shown in the Fig 11 and 12, respectively.

Moreover, Donepezil binds with one acyl binding site's amino acid residue such as Phe295 (Fig 10). Similarly, 'Roluperidone' bind with the same amino acid residues which are shown in the Fig 11. Although 'Napitane' do not bind with the acyl binding site of the active site pocket of hAChE, it binds with two amino acid residues of the

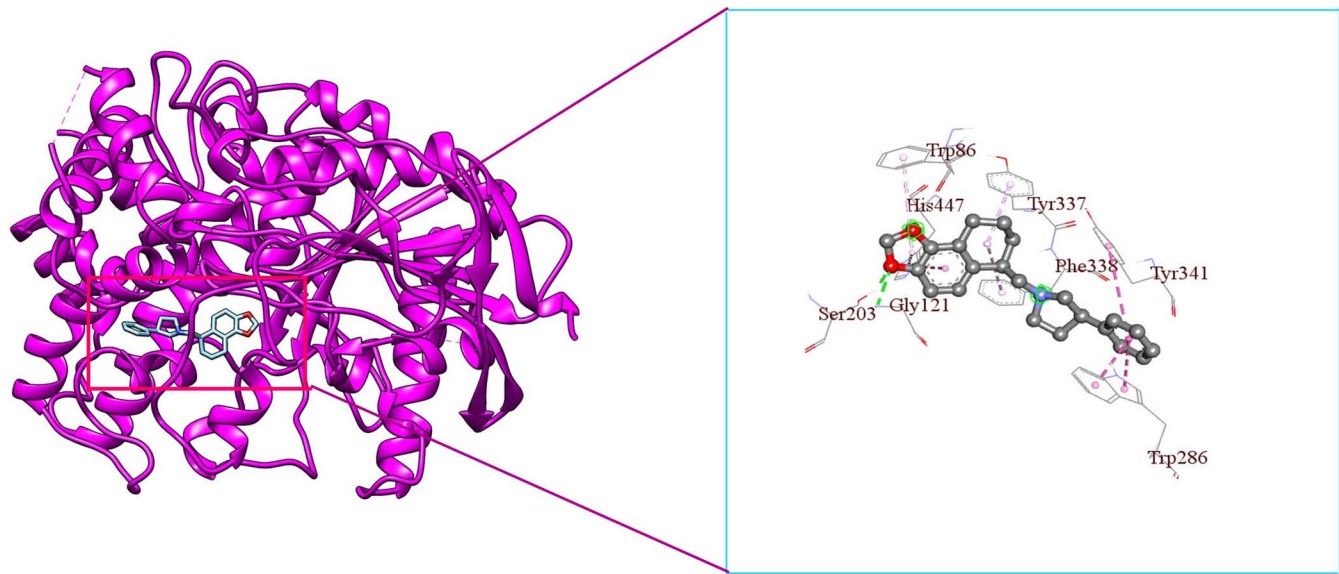

**Fig 12. 3D visualization of Napitane's binding mode and interaction with hAChE.**

acylation site of the active site pocket of hAChE such as Ser203 and His447 which are presented in Fig 12, and Fig 9. From the above discussion it could be summarized that 'Roluperidone' 'could be potential candidates for the inhibition of hAChE. However, we performed molecular dynamics simulation (MDS) of, hAChE- Roluperidone', and 'hAChE- Napitane' complexes to investigate their binding integrity to hAChE as well as compared to hAChE-Donepezil complex.

### 3.5  Molecular dynamics (MD) simulation

The dynamic behavior of protein-ligand complex is studied by MD simulation. Here, hAChE-Donepeil, hAChE-Napitane, hAChE-Roluperidone were studied in MD simulation for 100 ns by exploring various systemic and structural parameters described below.

**3.5.1 Root mean square deviation (RMSD) analysis.** Upon binding with ligand, the change of structural change is occurred in protein. Root-mean-square deviation (RMSD) parameter is used to evaluate the protein dynamic alteration [61].

The RMSD could be expressed by the following formula:

$$RMSD = \sqrt{\frac{\sum_{i=1}^{n} Ri * Ri}{n}}$$

where Ri is the vector linking the positions of atom i [of N atoms] in the reference snapshot [62].

The atomic RMSDs of Calpha [RMSDCa], backbone [RMSDBb] and all-heavy atom [RMSDAll] for both protein and the ligand of each complex for 100 ns are calculated and plotted in a time-dependent manner. Fig 13, Fig 14, Fig 15 demonstrate hAChE-Donepezil, hAChE-Napitane, and hAChE-Roluperidone RMSD respectively.

From Fig 13, it seems that the RMSD upon binding of the reference drug Donepezil with hAChE equilibrated from 25 to 80 ns. Upon binding of 'Roluperidone' and 'Napitane' with hAChE, the RMSD equilibrated from 25 to 90 ns, 24–75

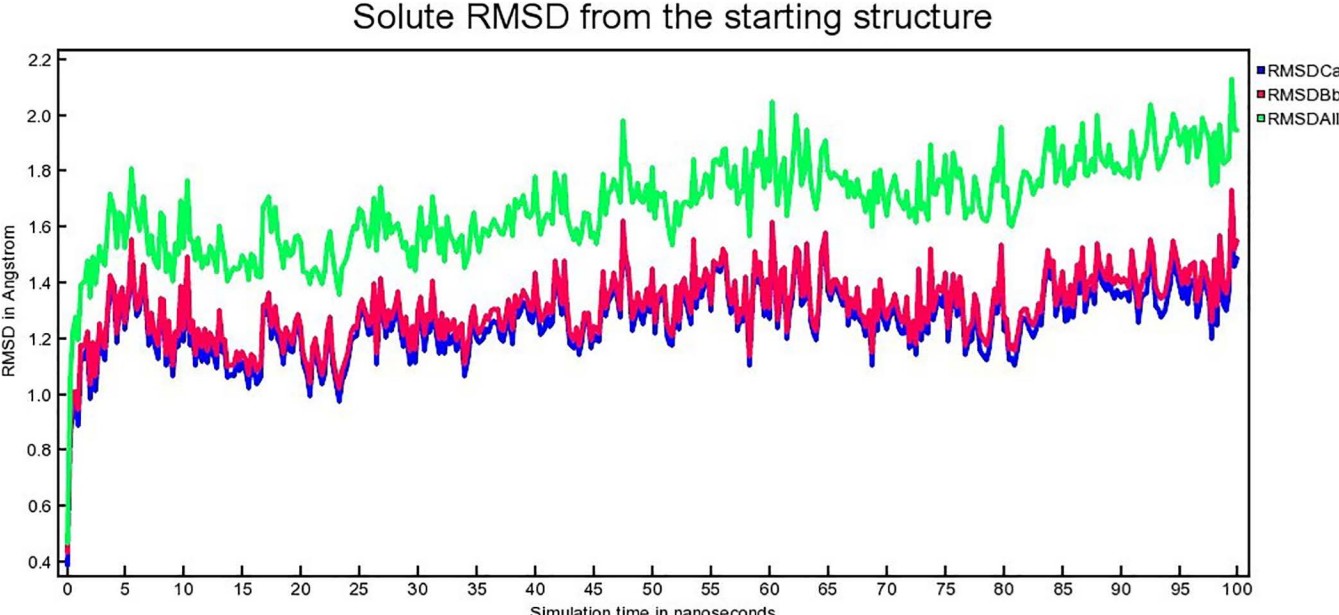

**Fig 13. The time series of the RMSD of of Calpha [RMSDCa], backbone [RMSDBb] and all-heavy atom [RMSDAll] of hAChE-Donepezil complex.**

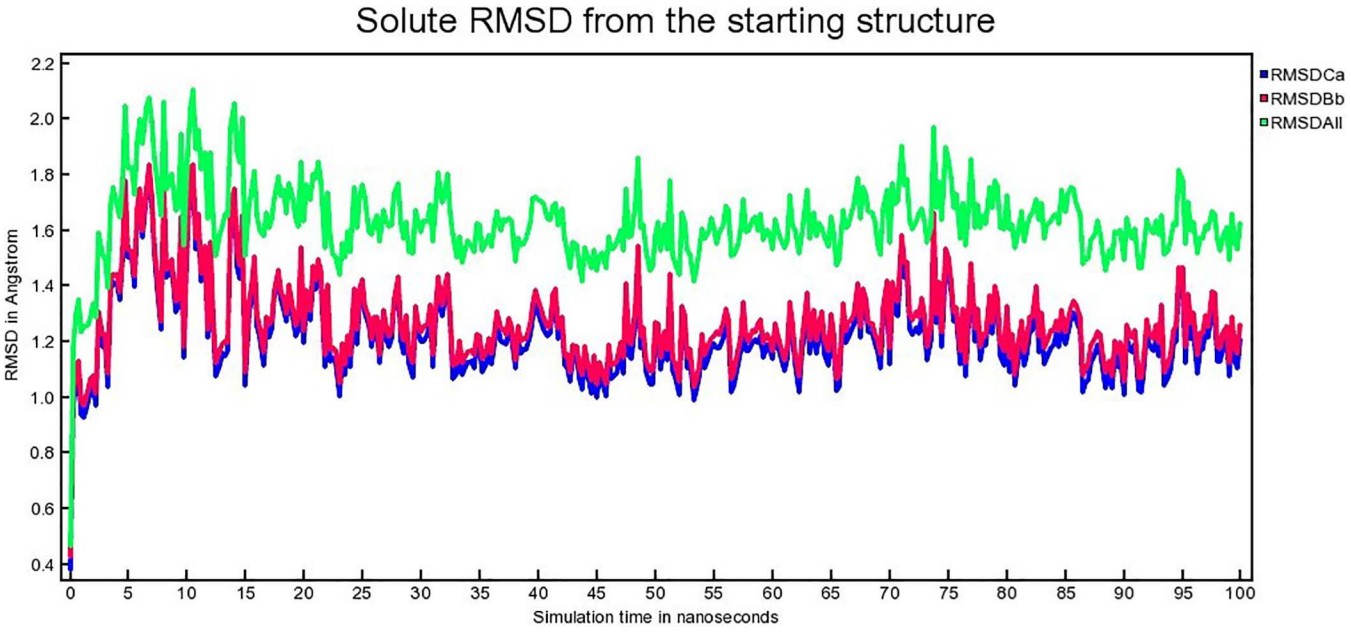

**Fig 14. The time series of the RMSD of of Calpha [RMSDCa], backbone [RMSDBb] and all-heavy atom [RMSDAll] of hAChE-Napitane complex.**

ns, respectively (Fig 14, Fig 15). From the above information, it is presumed that the least RMSD deviation occurred for 'Roluperidone', throughout the simulation than reference drugs, as well as Napitane. The smallest RMSD deviation denotes, the more stable a protein-ligand complex [63]. So, 'hAChE-Roluperidone' exhibits better structural integrity

## Solute RMSD from the starting structure

**Fig 15. The time series of the RMSD of of Calpha [RMSDCa], backbone [RMSDBb] and all-heavy atom [RMSDAll] of hAChE-Roluperidone complex.**

than the reference drug as well as hAChE-Napitane. The extent of inhibition of an inhibitor to the enzyme could be determined by calculating the structural integrity of the inhibitor-enzyme complex [64]. Since, the 'Roluperidone' shows comparatively better structural integrity than all other compounds binding with hAChE during MDS simulation, it may have a robust potential to inhibit hAChE.

**3.5.2 Radius of gyration.** The radius of gyration is presented by the following equation.

$$Radius_{gyr,Mass} = \sqrt{\frac{\sum_{i=1}^{n} Mass_i(\vec{R}_i - \vec{C})^2}{\sum_{i=1}^{N} Mass_i}}$$

Where, N = The total number of discrete elements (e.g., atoms, particles, or points) in the system being analyzed., Ri = The position vector of the i- element (its location in space), C = The center reference point. This is typically the center of mass of the entire system.

Radius of gyration connected to conformational state of protein. It denotes the compactness as well as folding of the protein [65]. The radius of gyration (Rg) for hAChE-Donepeil, hAChE-Napitane, and hAChE-Roluperidone complexes are depicted in Fig 16, Fig 17, Fig 18, respectively.

The hAChE-Donepezil complex shows a lower radius of gyration (Rg) value from approximately 3 ns to 75 ns, then its Rg value increased suddenly from 75 ns to 100 ns (Fig 16). hAChE- Napitane complex showed lower Rg value in the range of 15–40 ns, 50–70 ns (Fig 17). Throughout simulation, the hAChE-Roluperidone shows a steady state Rg value (Fig 18). The lower or steady-state Rg value for a protein-ligand complex indicates lower conformational changes of the protein in terms of more compactness [66]. So, it is concluded that hAChE showed more compactness upon 'Roluperidone' binding.

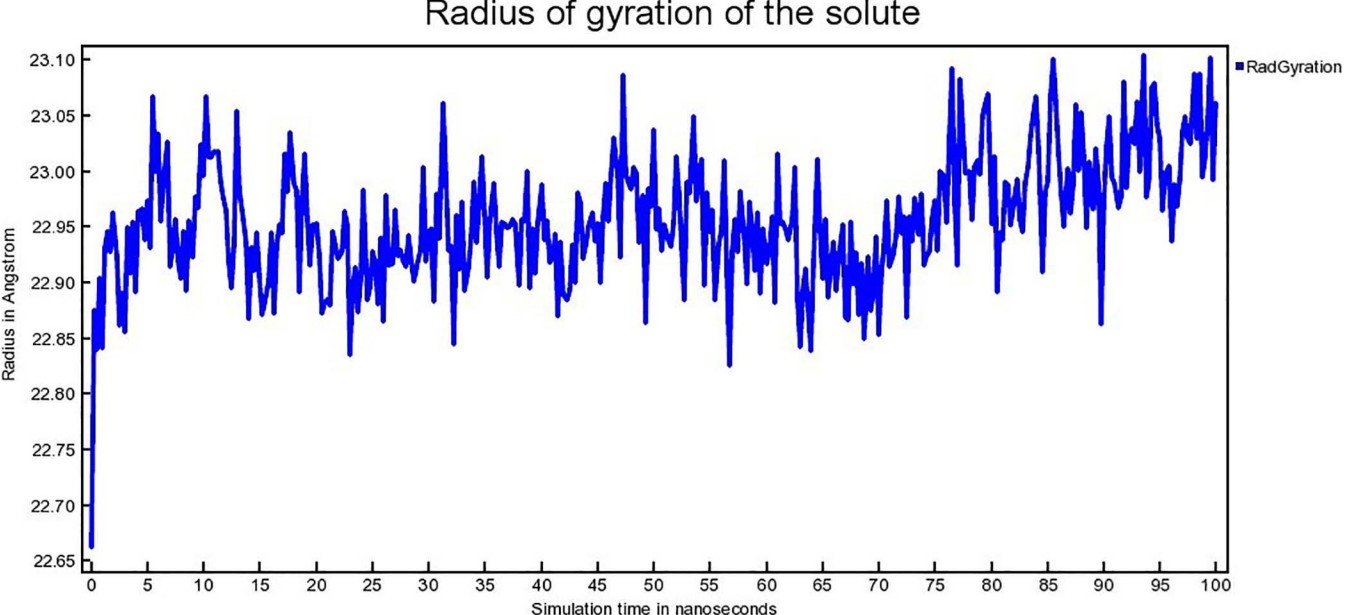

**Fig 16. The Radius of gyration (Rg) of hAChE- Donepezil complex.**

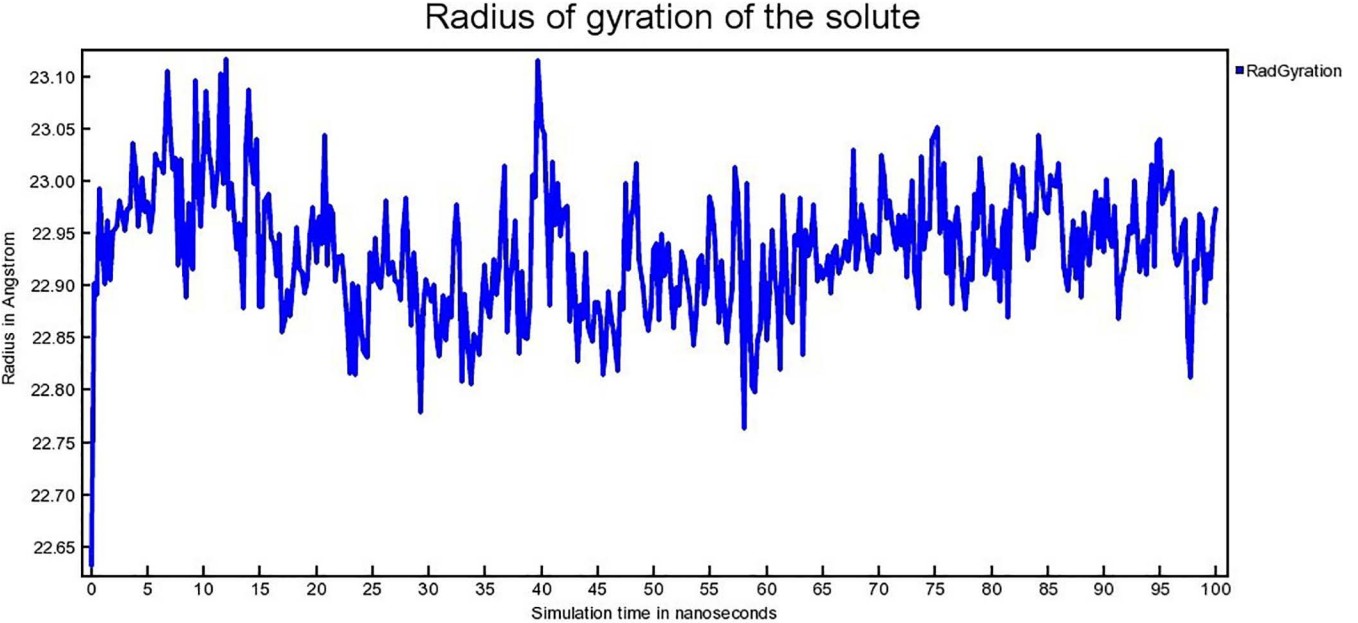

**Fig 17. The Radius of gyration (Rg) hAChE- Napitane complex.**

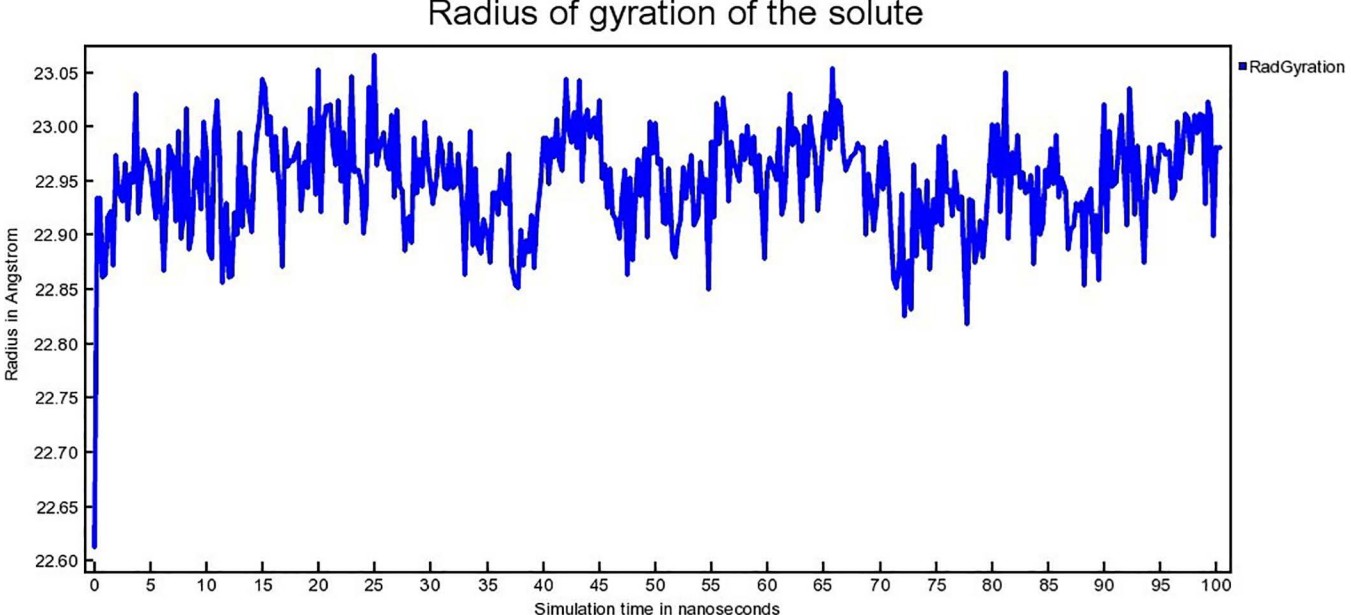

**Fig 18. The Radius of gyration (Rg) of hAChE- Roluperidone complex.**

**3.5.3 Root means square fluctuation (RMSF).** The RMSF is represented by the following equation.

$$RMSF_i = \sqrt{\sum_{j=1}^{3} \left( \frac{1}{N} \sum_{k=1}^{N} P_{ikj}^2 - \overline{P_{ij}}^2 \right)}$$

Where, N is the total number of frames or snapshots in the trajectory, k is the index for each frame/snapshot, running from 1 to N, P2 ij is the squared average position for particle i in dimension j

RMSF is used to study individual amino acid residues flexibility study(*i.e.,* measurement of the extent of particular amino acid residues fluctuations) during molecular dynamics. In Fig 19 especially represents the RMSF of PAS & CAS site amino acid residues of hAChE upon ligands bindings.

Here the RMSF of amino acid residues of CAS & PAS site of hAChE upon Donepezil binding is taken as a standard, the higher quantity of the PAS and CAS site residues site show lower RMSF upon 'Roluperidone' binding with hAChE (Fig 19). The lower RMSF indicates that the less conformational change occurs at the active site and PAS site upon 'Roluperidone' binding in term of hAChE-Roluperidone complex would be more stable during whole simulation. In the case of hAChE-Napitane complex, only one PAS site residue Trp286 had lower RMSF and other PAS site as well as CAS site residues show higher RMSF which denotes that Napitane-hAChE complex is not stable through the molecular simulation (Fig 19). Overall RMSF analysis suggested that 'Roluperidone' more strongly interact with PAS & CAS site residues of hAChE during whole 100 ns molecular dynamic simulation.

**3.5.4 Hydrogen bond analysis.** Here, Fig 20 to Fig 21, Fig 22 represent the quantity of H-bond formation in hAChE before ligands binding with it and hAChE-ligands complex throughout the complex.

By comparing Fig 20 with Fig 21, Fig 22, it is understood that the quantity of the h-bond increased upon Napitane binding in the range of 1–20 ns, 55–65 ns respectively. Since the hAChE-Napitane complex is stable in the mentioned ranges, it does not maintain it consistent integrity throughout simulation. By evaluating Fig 22, it is understood that

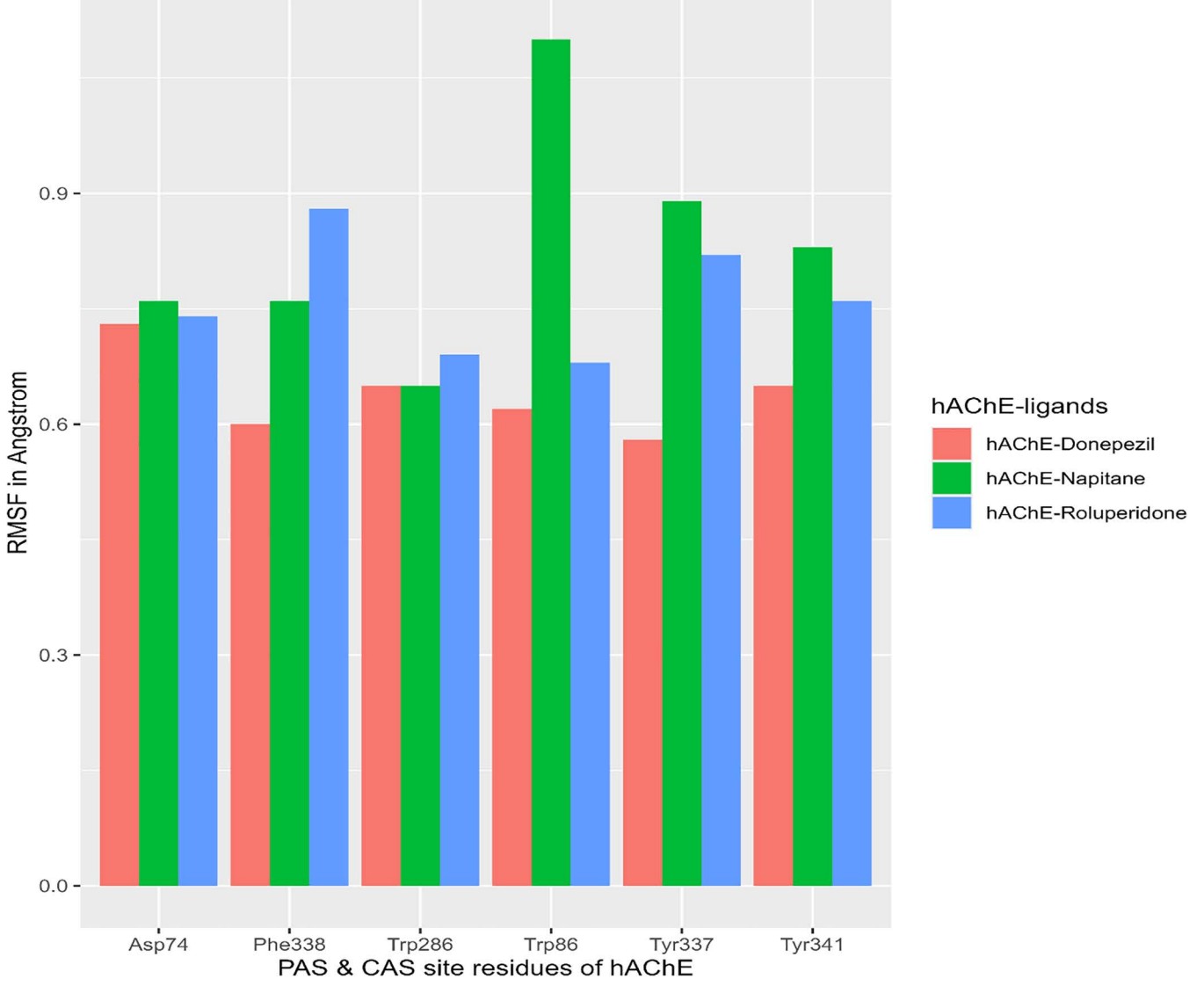

**Fig 19. The RMSF of the PAS and CAS site amino acid residues of hAChE in 100 ns MDS.** Here, the amino acid residues of catalytic anionic site (CAS) of hAChE are Trp86, Tyr337, Phe338, & the PAS site amino acid residues are Asp74, Trp286, and Tyr341.

upon Roluperidone binding with hAChE, the quantity of H-Bond formation are geared up gradually from 5 to 45 ns, then slightly decrease then a steady state level is obtained, which implies that the hAChE-Roluperidone complex is a more stable form. Because a higher number of hydrogen bond formations between a ligand and a protein generally signifies a stronger and more favorable interaction. From the overall H-bonding analysis, it is understood that among all the complexes, Roluperidone-hAChE complex shows better structural integrity.

## 4. Conclusion

The quantity of available marketed drugs for AD is not sufficient. Hence, there is an urgent need to add a substantial quantity of drugs to the AD drug discovery pipeline. Drug repurposing could be a prominent approach in this regard.

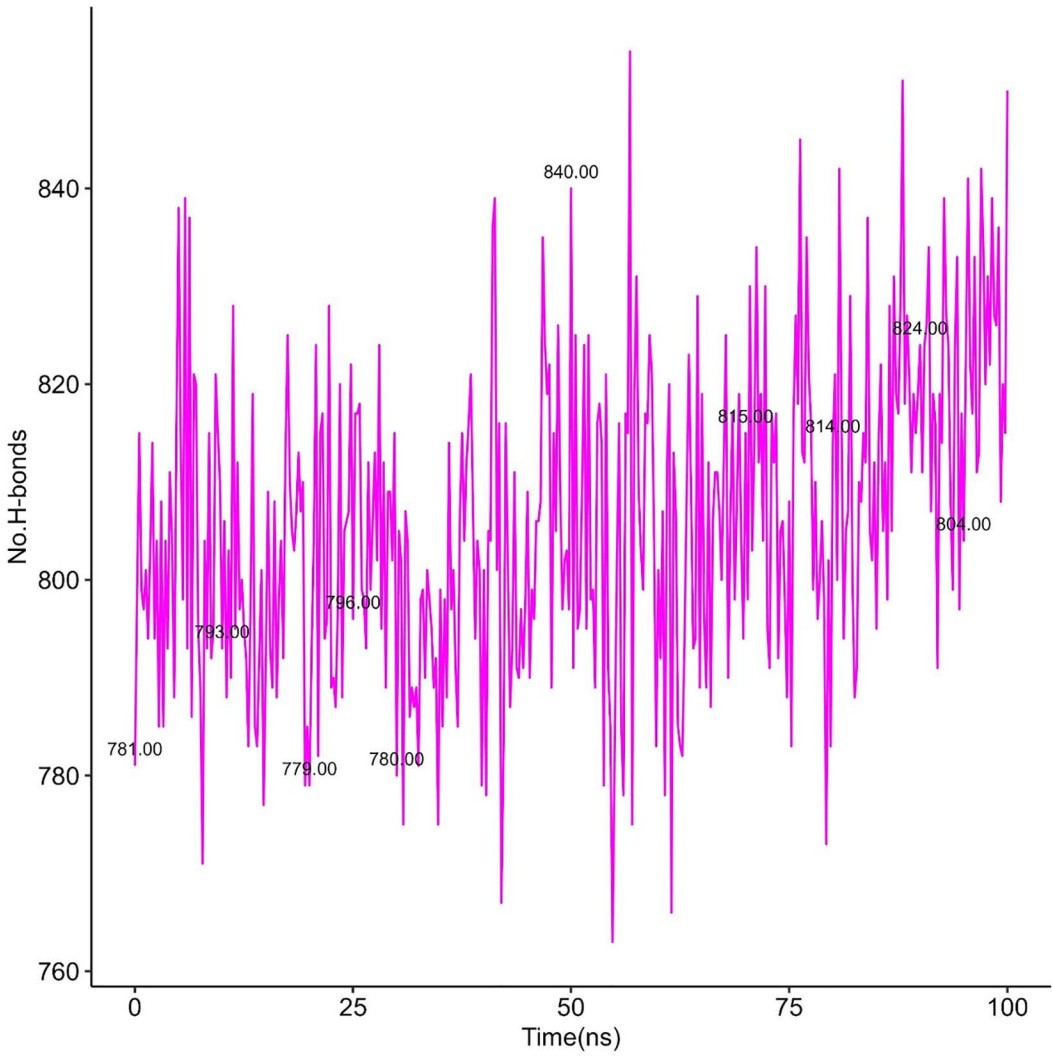

**Fig 20. H-bond in hAChE-Donepezil complex.**

This study is conducted to identify novel repurposing candidates for AD through machine learning (ML)algorithms, chemoinformatic analysis, molecular docking, and molecular dynamics simulation (MDS). Among the generated machine learning model, the SVM showed 81% accuracy during 5-fold cross-validation with an AUC value of 0.81. Due to high accuracy, the SMV model was applied to find out novel repurposing potential compounds for AD from 500 CNS active compounds & among these compounds 60 compounds were predicted as AD repurposing potential. Then chemoinformatic analysis revealed that 9 compounds from 60 compounds were structurally more similar to the reference drug Donepezil. After the molecular docking performance of the 9 compounds, 2 compounds showed higher binding affinity - 'Roluperidone' had a binding affinity of −12 kcal/mol, and 'Napitane' had a binding affinity of −11.9 kcal/mol whereas Donepezil had binding affinity of −11.8 kcal/mol. All the compounds bind with crucial amino acid residues of the PAS and CAS site of hAChE. Molecular Dynamics Simulations revealed that 'Roluperi-done' showed equilibrium RMSD from 25 to 90 ns which is better than any other compounds presented in this study.

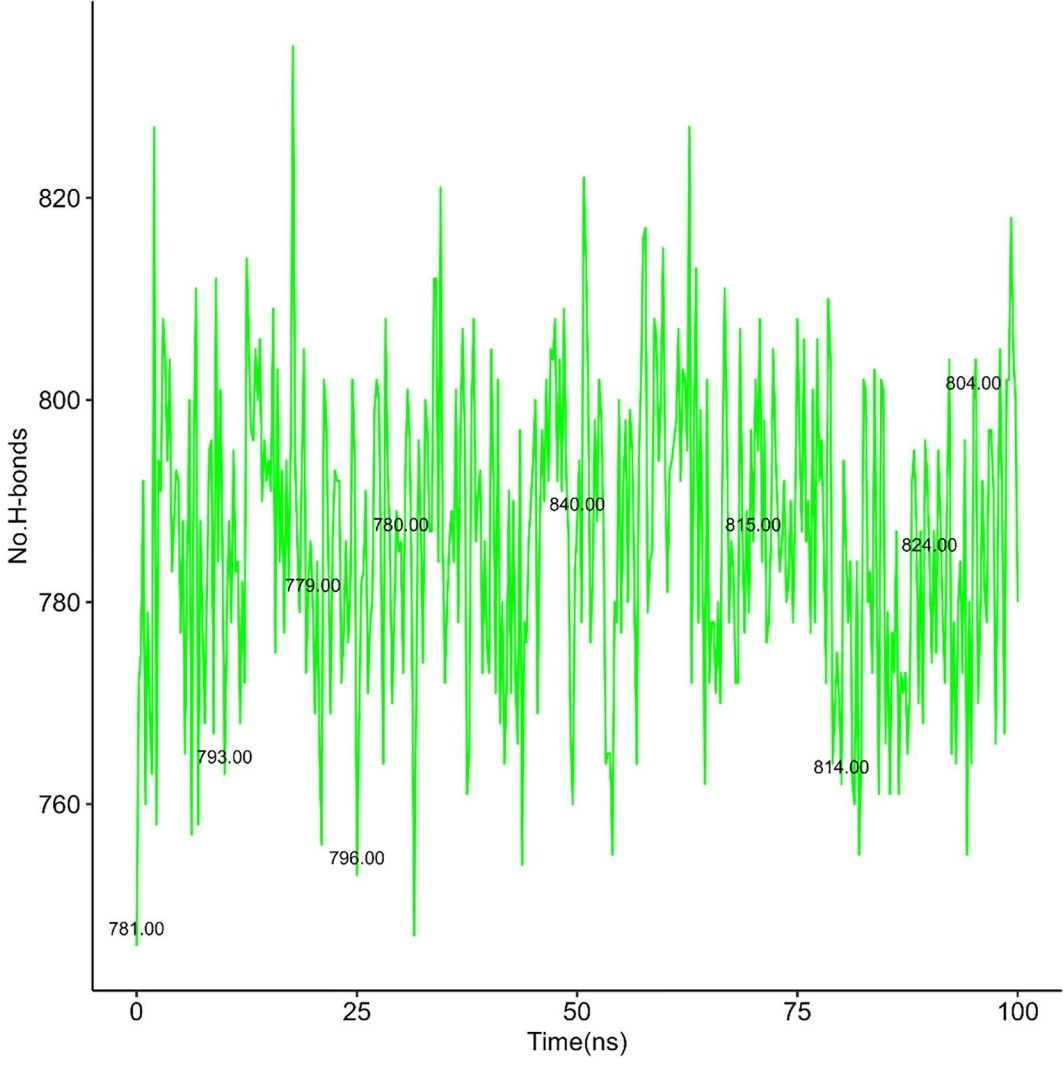

**Fig 21. H-bond in hAChE-Napitane complex.**

Further, radius of gyration, RMSF, and H-bond analysis support this result and finally it is understood that Roluperidone has comparatively stable binding potential toward the human acetylcholinesterase (hAChE). We suggest that 'Roluperidone' could be further explored in *in-vitro* and *in-vivo* settings (*i.e.* Ellman's method, *etc.*) to develop as a repurposing candidate for AD.

## 5. About the potential repurposing candidate: 'Roluperidone'

Roluperidone is an investigational drug. To treat schizophrenia, Rlouperidone was investigated in phase 2b and 3 clinical trials. It has an affinity for the $5HT_{2A}$ receptor [67,68]. Moreover, it blocks $sigma_2$ receptor, which is involved in movement control, memory, etc. Since Roluperidone has multiple target interactions, it may have the potential to inhibit hAChE. However, this in silico study is not sufficient. Therefore, *in-vitro* and *in-vivo* experiments are needed for that.

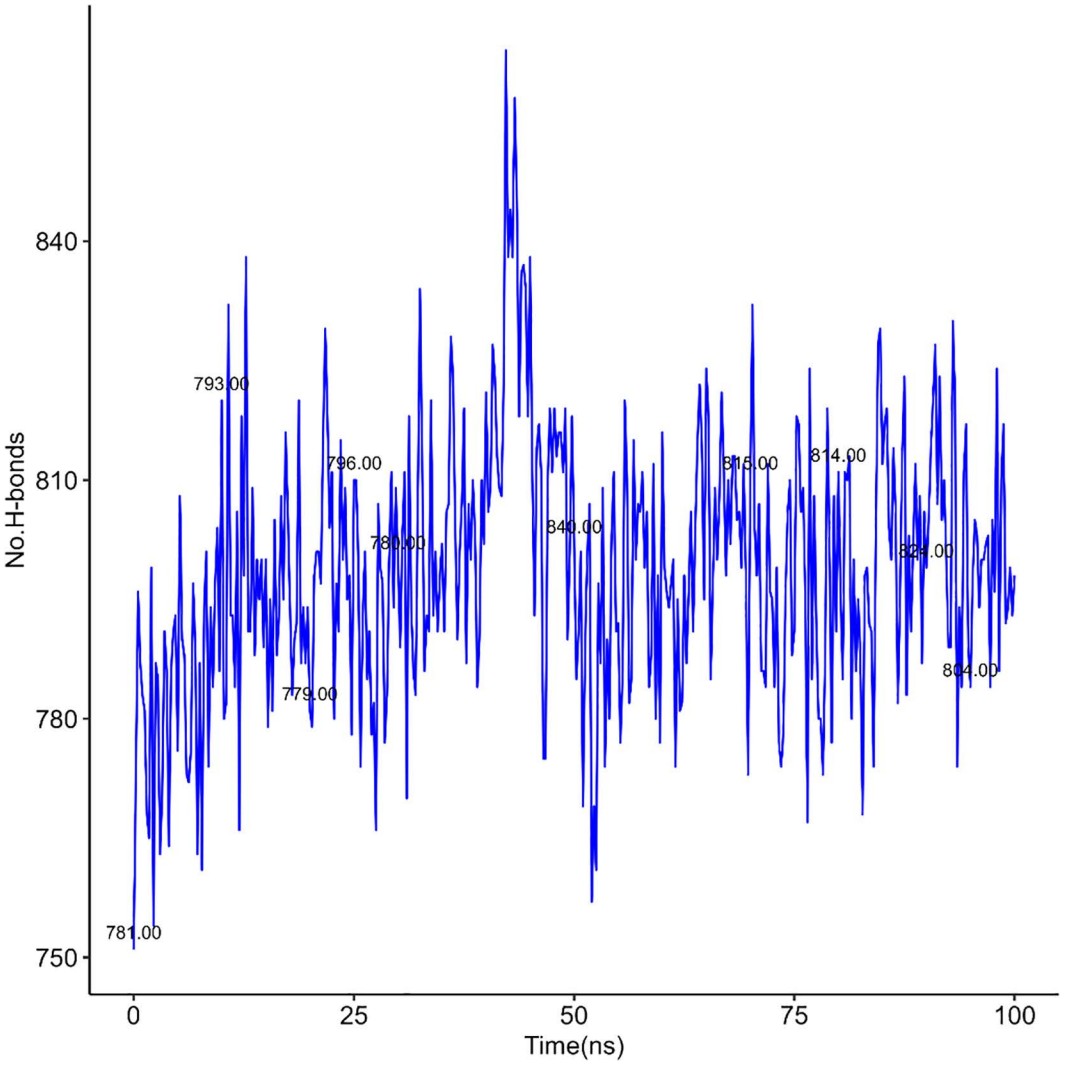

**Fig 22. H-bond in hAChE-Roluperidone complex.**

## Supporting information

**S1 File. AD Repurposing Candidate Enlistment.**
(XLSX)

**S2 File. Training Building Dataset.**
(ZIP)

**S3 File. Python script for machine learning model generation.**
(DOCX)

**S4 File. Rcdk script for fingerprint calculation.**
(TXT)

**S5 File. Similarity scores.**
(XLSX)

**S6 File. Binding Affinity.**
(ZIP)

**S7 File. Molecular Dynamic Simulation.**
(ZIP)

## Author contributions

**Conceptualization:** Rehnuma Tanjin, Md. Al-Amin, Jannatul Mawa Etee, Ayesha Siddika, Saiful Islam Mahi, Ahmadullah Siddiki, Sharmin Nur Toma, Nafisa Akter, Md. Faruk Hossen, Md. Helal Uddin, Neelima Akhter Bristy, Samira Idris Mowlee, Elmu Kabir Rafa.

**Data curation:** Md. Al-Amin, Ahmadullah Siddiki, Nafisa Akter, Md. Faruk Hossen, Md. Helal Uddin, Neelima Akhter Bristy, Samira Idris Mowlee, Elmu Kabir Rafa.

**Formal analysis:** Rehnuma Tanjin, Md. Al-Amin, Jannatul Mawa Etee, Ayesha Siddika, Nafisa Akter, Md. Helal Uddin.

**Funding acquisition:** Rehnuma Tanjin, Md. Al-Amin, Jannatul Mawa Etee, Ayesha Siddika, Saiful Islam Mahi, Ahmadullah Siddiki, Sharmin Nur Toma, Nafisa Akter.

**Investigation:** Jannatul Mawa Etee, Ayesha Siddika, Saiful Islam Mahi, Ahmadullah Siddiki, Sharmin Nur Toma, Nafisa Akter.

**Methodology:** Md. Al-Amin.

**Project administration:** Rehnuma Tanjin.

**Resources:** Md. Al-Amin, Sharmin Nur Toma, Nafisa Akter.

**Software:** Md. Al-Amin, Sharmin Nur Toma.

**Supervision:** Rehnuma Tanjin, Saiful Islam Mahi, Ahmadullah Siddiki.

**Validation:** Rehnuma Tanjin.

**Visualization:** Rehnuma Tanjin, Md. Al-Amin, Jannatul Mawa Etee, Ayesha Siddika, Saiful Islam Mahi, Ahmadullah Siddiki, Sharmin Nur Toma, Nafisa Akter, Md. Helal Uddin.

**Writing – original draft:** Rehnuma Tanjin, Md. Al-Amin, Jannatul Mawa Etee, Ayesha Siddika, Nafisa Akter.

**Writing – review & editing:** Rehnuma Tanjin, Md. Al-Amin, Jannatul Mawa Etee, Ayesha Siddika, Nafisa Akter, Md. Faruk Hossen, Md. Helal Uddin, Neelima Akhter Bristy, Samira Idris Mowlee, Elmu Kabir Rafa.

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
