## [Decision Letter · Decision Letter 0]

27 Oct 2025

PONE-D-25-42057An Evaluation of Roluperidone as a Promising Repurposing Candidate for Alzheimer’s DiseasePLOS ONE

Dear Dr. Al-Amin,

Thank you for submitting your manuscript to PLOS ONE. After careful consideration, we feel that it has merit but does not fully meet PLOS ONE’s publication criteria as it currently stands. Therefore, we invite you to submit a revised version of the manuscript that addresses the points raised during the review process.

Researchers performed an extensive virtual screening to reconsider different drugs likely linked to Alzheimer's disease. They refined these to nine molecules, ultimately identifying Roluperidone as a prospective candidate for Alzheimer's disease treatment. This screening technique presented literature examples of both virtual screening and the in vitro experimental efficacy of particular drugs in preventing Alzheimer's disease. At this point, more extensive screening may be performed, and research previously assessed in vitro or in vivo against AD could be examined and integrated into the article.==============================

We look forward to receiving your revised manuscript.

Kind regards,

Zeynep Ozdemir

Academic Editor

PLOS ONE

Journal Requirements:

3. Please amend the manuscript submission data (via Edit Submission) to include author Neelima Akhter Bristy

4. Please ensure that you refer to Figures 1 and 21 in your text as, if accepted, production will need this reference to link the reader to the figures.

5. Please upload a copy of Supporting Information files which you refer to in your text on page 29.

Additional Editor Comments:

Researchers performed an extensive virtual screening to reconsider different drugs likely linked to Alzheimer's disease. They refined these to nine molecules, ultimately identifying Roluperidone as a prospective candidate for Alzheimer's disease treatment. This screening technique presented literature examples of both virtual screening and the in vitro experimental efficacy of particular drugs in preventing Alzheimer's disease. At this point, more extensive screening may be performed, and research previously assessed in vitro or in vivo against AD could be examined and integrated into the article. Furthermore, necessary revisions must be implemented in accordance with the referee's recommendations.

Reviewers' comments:

Reviewer's Responses to Questions

**Comments to the Author**

1. Is the manuscript technically sound, and do the data support the conclusions?

Reviewer #1: Yes

2. Has the statistical analysis been performed appropriately and rigorously? 

Reviewer #1: Yes

3. Have the authors made all data underlying the findings in their manuscript fully available?

Reviewer #1: Yes

4. Is the manuscript presented in an intelligible fashion and written in standard English?

Reviewer #1: No

5. Review Comments to the Author

Reviewer #1: The manuscript entitled “An Evaluation of Roluperidone as a Promising Repurposing Candidate for Alzheimer’s Disease” has been reviewed.

The manuscript presents a well-designed study in the field of computational chemistry. I recommend publishing this manuscript after minor revision:

- The manuscript should be checked for spelling errors, as there are many misspellings and errors.

-In the introduction, the computer-aided drug design section was briefly explained with a very good introduction, however, a figure explaining the structure and ligand-based design could be added.

- Abbreviations should be written according to the same standards.

Abbreviations that appear as "Ach" should be corrected to "ACh."

IC50 spellings should be written according to the rules.

Words that should be italicized should be corrected. (For example, in vitro, etc.)

- Figure and supporting information citations must be written correctly. In many places, citations were indicated as Figure X and SX.

- It can be explained in detail which tools they use in fingerprint calculations.

- The docking method was quite simple and straightforward. However, the 25x25x25 area determined during the docking process should be explained in detail.

- The reasons for selecting these drugs based on the similarity graph in 3.3.1 should be detailed. The basis for the machine learning elimination, which parameters stood out, and so on, should be specified.

- Bond lengths are also given in Table 3. The study focuses on other important aspects, but bond lengths are not mentioned. Are physicochemical parameters used in the calculation, or where are they important for comparison? If they are not important, what is the purpose of including them on the table?

- The molecular dynamics studies were well interpreted. The tools used were different, but are the underlying equations in their calculations the same? This needs to be explained.

6. PLOS authors have the option to publish the peer review history of their article (what does this mean? ). If published, this will include your full peer review and any attached files.

**Do you want your identity to be public for this peer review?** For information about this choice, including consent withdrawal, please see our Privacy Policy .

Reviewer #1: No

---

## [Author Response · Author response to Decision Letter 1]

12 Nov 2025

Dear Editor & Reviewers,

We sincerely appreciate the time and effort you invested in reviewing our manuscript. We have carefully addressed all your instructions and comments. A detailed list of the corrections implemented is presented in the right column of the following table.

-Md. Al-Amin (Correspondent Author)

Serial No. Editor/Reviewers' comments Responses

1. Researchers performed an extensive virtual screening to reconsider different drugs likely linked to Alzheimer's disease. They refined these to nine molecules, ultimately identifying Roluperidone as a prospective candidate for Alzheimer's disease treatment. This screening technique presented literature examples of both virtual screening and the in vitro experimental efficacy of particular drugs in preventing Alzheimer's disease. At this point, more extensive screening may be performed, and research previously assessed in vitro or in vivo against AD could be examined and integrated into the article. Furthermore, necessary revisions must be implemented in accordance with the referee's recommendations. Thank you for your insightful suggestion. We have added several citations containing previously assessed in vitro or in vivo against AD could be examined and integrated into the article.in the revised section of the manuscript

2. In the introduction, the computer-aided drug design section was briefly explained with a very good introduction; however, a figure explaining the structure and ligand-based design could be added.

Thank you for your insightful instruction. We have added a figure in that section that explains the structure and ligand-based design.

3. Abbreviations should be written according to the same standards. Thank you. Abbreviations have been written according to the same standards.

4. Abbreviations that appear as "Ach" should be corrected to "ACh." Thank you. "Ach" has been corrected to "ACh."

5. IC50 spellings should be written according to the rules. Thank you. The IC50 spellings have been corrected.

6. Words that should be italicized should be corrected. (For example, in vitro, etc.)

Thank you. It has been corrected

7. Figure and supporting information citations must be written correctly. In many places, citations were indicated as Figure X and SX.

Thank you for the careful revision. We have corrected it.

8. It can be explained in detail which tools they use in fingerprint calculations. Thank you. We have explained in detail which tools they use in fingerprint calculations in the revised section of the manuscript.

9. The docking method was quite simple and straightforward. However, the 25x25x25 area determined during the docking process should be explained in detail. Thank you. We have elaborated this section in the revised manuscript.

10. Bond lengths are also given in Table 3. The study focuses on other important aspects, but bond lengths are not mentioned. Are physicochemical parameters used in the calculation, or where are they important for comparison? Thank you. We have corrected it. We have added a figure instead of the table where bond lengths are depicted in the revised manuscript.

11. The molecular dynamics studies were well interpreted. The tools used were different, but are the underlying equations in their calculations the same? This needs to be explained. Thank you. We have revised the molecular dynamics studies. The equations for calculating the molecular dynamic parameter are given.

12. The reasons for selecting these drugs based on the similarity graph in 3.3.1 should be detailed.

Thank you. We described it in the manuscript.

13. The basis for the machine learning elimination, which parameters stood out, and so on, should be specified. Thank you for your careful concern for the manuscript. Initially, we performed similarity analysis before machine learning. But, later, according to the project administration instruction, we conducted machine learning before similarity analysis. The previous section of similarity analysis was not correct unconsciously. We did not eliminate the machine learning. We made a mistake. We have felt sorrow for that. However, we have corrected these in the revised section.

---

## [Decision Letter · Decision Letter 1]

20 Nov 2025

An Evaluation of Roluperidone as a Promising Repurposing Candidate for Alzheimer’s Disease: A Computational Investigation

PONE-D-25-42057R1

Dear Dr. Al-Amin,

We’re pleased to inform you that your manuscript has been judged scientifically suitable for publication and will be formally accepted for publication once it meets all outstanding technical requirements.

Kind regards,

Zeynep Ozdemir

Academic Editor

PLOS ONE

Additional Editor Comments (optional):

All comments have been addressed.

Reviewers' comments:

Reviewer's Responses to Questions

**Comments to the Author**

1. If the authors have adequately addressed your comments raised in a previous round of review and you feel that this manuscript is now acceptable for publication, you may indicate that here to bypass the “Comments to the Author” section, enter your conflict of interest statement in the “Confidential to Editor” section, and submit your "Accept" recommendation.

Reviewer #1: All comments have been addressed

2. Is the manuscript technically sound, and do the data support the conclusions?

Reviewer #1: Yes

3. Has the statistical analysis been performed appropriately and rigorously? 

Reviewer #1: Yes

4. Have the authors made all data underlying the findings in their manuscript fully available?

Reviewer #1: (No Response)

5. Is the manuscript presented in an intelligible fashion and written in standard English?

Reviewer #1: Yes

6. Review Comments to the Author

Reviewer #1: (No Response)

7. PLOS authors have the option to publish the peer review history of their article (what does this mean? ). If published, this will include your full peer review and any attached files.

**Do you want your identity to be public for this peer review?** For information about this choice, including consent withdrawal, please see our Privacy Policy .

Reviewer #1: No

---

## [Editor Report · Acceptance letter]

PONE-D-25-42057R1

PLOS One

Dear Dr. Al-Amin,

I'm pleased to inform you that your manuscript has been deemed suitable for publication in PLOS One. Congratulations! Your manuscript is now being handed over to our production team.

Kind regards,

on behalf of

Professor Zeynep Ozdemir

Academic Editor

PLOS One